# Selective STAT3 Allosteric Inhibitors HCB-5300 and HCB-5400 Alleviate Dextran Sulfate Sodium-Induced Ulcerative Colitis in Mice

**DOI:** 10.3390/ijms262411981

**Published:** 2025-12-12

**Authors:** Wook-Young Baek, Ji-Won Kim, So-Won Park, Nan Kim, Sun-Gyo Lim, Chang-Hee Suh

**Affiliations:** 1Department of Rheumatology, Ajou University School of Medicine, Suwon 16499, Republic of Korea; arikato83@naver.com (W.-Y.B.); jwk722@naver.com (J.-W.K.); 2Department of Molecular Science and Technology, Ajou University, Suwon 16499, Republic of Korea; sowon9413@naver.com; 3Huchembio, Room 807, Golden I-Tower, 570-6 Dongtan-giheung-ro, Hwaseong 18469, Republic of Korea; nan4471@huchembio.com; 4Department of Gastroenterology, Ajou University School of Medicine, Suwon 16499, Republic of Korea; mdlsk75@ajou.ac.kr

**Keywords:** inflammatory bowel disease, STAT3 inhibitor, ulcerative colitis, anti-inflammatory, DSS-induced colitis

## Abstract

Inflammatory bowel disease (IBD) is a chronic inflammatory disorder of the gastrointestinal tract, marked by persistent mucosal inflammation and structural damage. We evaluated the efficacy of HCB-5300 and HCB-5400, novel selective STAT3 allosteric inhibitors, in a mouse model of dextran sulfate sodium (DSS)-induced ulcerative colitis. Colitis was induced in C57BL/6 mice using 3% DSS in water for 5 d. HCB-5300 (25 mg/kg) or HCB-5400 (12.5 mg/kg) was administered orally during induction. Disease progression was assessed using the disease activity index (DAI), considering body weight, stool consistency, and rectal bleeding. Colon length and histopathological analyses were used to evaluate mucosal integrity and inflammatory damage. Interleukin (IL)-6 levels were quantified using enzyme-linked immunosorbent assay, and kidney pathology was assessed for systemic effects. HCB-5300 and HCB-5400 significantly mitigated DSS-induced colitis, as evidenced by reduced body weight loss, improved DAI scores, preserved colon length, and decreased mucosal damage and inflammation in the treated mice. IL-6 levels were significantly lower in both treatment groups, indicating effective STAT3 inhibition. HCB-5400 exhibited superior efficacy for most parameters. HCB-5300 and HCB-5400 are potent and selective STAT3 allosteric inhibitors with notable anti-inflammatory effects. HCB-5400’s efficacy underscores its potential as a therapeutic candidate for managing inflammatory flares in IBD.

## 1. Introduction

Inflammatory bowel disease (IBD), including ulcerative colitis (UC) and Crohn’s disease (CD), is a chronic, relapsing inflammatory disorder of the gastrointestinal tract characterized by mucosal inflammation, epithelial barrier dysfunction, and dysregulated immune responses [1]. The etiology of IBD is multifactorial, involving genetic susceptibility, environmental factors, alterations in the gut microbiota, and immune dysregulation [2]. Over the past decades, IBD incidence has significantly increased worldwide, particularly in newly industrialized countries, highlighting the urgent need for effective and safe therapeutic interventions [3].

Current therapeutic approaches for IBD aim to suppress inflammation and restore intestinal homeostasis. Conventional treatments include 5-aminosalicylic acid, corticosteroids, immunosuppressants (e.g., azathioprine, methotrexate), and biologics targeting tumor necrosis factor-alpha (TNF-α), interleukin (IL)-12/23, and integrins [4,5]. Although biologics such as infliximab and vedolizumab have revolutionized IBD treatment, approximately 30–40% of patients exhibit primary non-response or develop secondary resistance over time [6]. Moreover, these treatments are associated with an increased risk of infections, malignancies, and loss of efficacy due to anti-drug antibody formation [7].

Janus kinase (JAK) inhibitors, such as tofacitinib and upadacitinib, have emerged as alternative small-molecule treatments for IBD [8]. These agents offer broad immunomodulatory effects by inhibiting several cytokine pathways [8]. However, concerns regarding their long-term safety profiles persist, particularly regarding the risks of thromboembolism, infections, and dyslipidemia [9]. For instance, the risk of herpes zoster infection has been reported to increase in patients receiving JAK inhibitors [10].

Considering these therapeutic limitations, the selective inhibition of signal transducer and activator of transcription 3 (STAT3) has emerged as a compelling, mechanistically driven strategy for targeted intervention in IBD. STAT3 is pivotal in IBD pathogenesis as it regulates inflammatory cytokines (e.g., IL-6, IL-23, TNF-α), maintains intestinal epithelial integrity, and influences immune cell differentiation. Unlike JAK inhibitors, which broadly suppress the JAK-STAT signaling pathway, STAT3-selective inhibitors provide a more targeted approach by mitigating excessive inflammation while minimizing systemic immunosuppression [11].

The dextran sulfate sodium (DSS)-induced colitis model is one of the most widely used experimental models for studying the pathophysiology of IBD and evaluating potential therapeutic agents. DSS disrupts the intestinal epithelial barrier, allowing luminal antigens to penetrate the mucosa and trigger innate and adaptive immune responses, mimicking the key features of human UC [12]. This model allows assessment of disease severity based on body weight loss, disease activity index (DAI), colon shortening, histopathological changes, and cytokine dysregulation [13].

In this study, we evaluated the therapeutic potential of two novel selective STAT3 small-molecule inhibitors, HCB-5300 and HCB-5400, in a DSS-induced colitis model. Considering that a more targeted approach than broad JAK inhibition may provide a better safety profile, these compounds were designed to modulate STAT3-mediated intestinal inflammation and cytokine signaling pathways specifically. We hypothesized that the selective inhibition of STAT3 by HCB-5300 and HCB-5400 would mitigate colonic injury, suppress pro-inflammatory cytokine expression (e.g., IL-6), and improve histopathological outcomes in DSS-induced colitis. To systematically assess their efficacy, we analyzed key clinical disease parameters, including body weight changes and DAI, as well as histopathological alterations and cytokine expression profiles. Collectively, our findings provide novel mechanistic insights and support the potential therapeutic utility of HCB-5300 and HCB-5400 as promising targeted candidates for the treatment of IBD.

## 2. Results

### 2.1. HCB Compounds Selectively Inhibit STAT3 Transcriptional Activity Without Affecting Cell Viability

To assess the selectivity of the orally available small-molecule inhibitors HCB-5300 and HCB-5400, we investigated their effects on the transcriptional activity of different STAT isoforms using a luciferase reporter assay. Both compounds demonstrated potent and highly selective inhibition of STAT3 transcriptional activity, with half-maximal inhibitory concentration (IC_50_) values of 0.06060 and 0.01537 μM for HCB-5300 and HCB-5400, respectively (Figure 1B). These data identify HCB-5300 as an initial lead compound and HCB-5400 as a lead-optimized analog. The IC_50_ values indicate that HCB-5400 is approximately 4-fold more potent than HCB-5300 in its direct inhibition of STAT3. This clear difference in in vitro potency provides the primary mechanistic rationale for the superior in vivo efficacy of HCB-5400 that is described in subsequent sections.

In contrast, the inhibitors did not affect the transcriptional activity of other STAT isoforms, such as STAT1, even at concentrations up to 1 μM. The calculated IC_50_ values against STAT1 were 80.59 μM for HCB-5300 and 16.75 μM for HCB-5400, highlighting their significant selectivity toward STAT3 (Figure 1A). As a control, the JAK inhibitor deucravacitinib broadly reduced the transcriptional activity of all tested STAT isoforms, with IC_50_ values of 0.2598 μM for STAT3 and 2.779 μM for STAT1 (Figure 1A,B).

Furthermore, the cytotoxicity of HCB compounds was evaluated in parallel with luciferase reporter assays. No reduction in the viability of either HeLa or HEK293 cells was observed at concentrations of up to 0.5 μM. The HCB compounds maintained high cell viability at much higher concentrations, confirming that the observed inhibition of STAT3 activity was not due to nonspecific cytotoxicity (Figure 1C,D).

The doses for subsequent in vivo efficacy studies were determined based on these in vitro mechanistic data (i.e., STAT3 inhibitory IC_50_) and the pharmacokinetic profiles of the compounds. The effective dose range was established by benchmarking against both established STAT3 selective inhibitors and broader-acting JAK inhibitors. The oral doses used in the DSS-induced colitis model were selected on the basis of the in vitro STAT3 inhibitory IC50 values and preliminary pharmacokinetic profiles of HCB-5300 and HCB-5400, which indicated that 25 mg/kg and 12.5 mg/kg, respectively, achieved plasma exposures expected to exceed the cellular IC50 over the dosing interval (Appendix A).

In addition, we confirmed that HCB-5300, a representative compound of this STAT3 inhibitor series, functions as a selective allosteric inhibitor of STAT3 through mechanistic studies, particularly JAK inhibition profiling and biotin pull-down assays, the results of which are comprehensively summarized in Appendix A.

### 2.2. Evaluation of Anti-Inflammatory Activity (IL-6 and IL-17A Inhibitory Activity)

IL-6 is a cytokine involved in innate and adaptive immunity, and is produced by various cell types, including lymphoid or mononuclear cells, B cells, T cells, vascular endothelial cells, fibroblasts, and keratinocytes. It also induces the expression of other inflammatory cytokines, including IL-6, and inflammatory mediators, such as nitric oxide (NO), via inducible nitric oxide synthase in macrophages. In this study, Raw264.7 cells were treated with HCB compounds at various concentrations (0.0003–30 μM), and IL-6 expression levels were measured. HCB-5400 exhibited a dose-dependent inhibitory effect, exceeding 80% inhibition observed at concentrations above 0.3 μM (Figure 2A). The (−) lipopolysaccharide (LPS)-treated group was used as a reference for 100% inhibition, whereas the reference compound dexamethasone (10 μM) exhibited 94.94% inhibition (Figure 2A).

IL-17 is a cytokine responsible for defense against bacteria and fungi along with modulating inflammatory responses. However, excessive IL-17 production is associated with the development of chronic inflammatory and autoimmune diseases, such as psoriasis, psoriatic arthritis, rheumatoid arthritis, and ankylosing spondylitis.

Subsequently, HaCaT cells were treated with HCB compounds at various concentrations (0.0003–30 μM). IL-17 measurements revealed that HCB-5400 inhibited IL-17 expression in a dose-dependent manner, with >80% inhibition at concentrations > 0.3 μM. This inhibition profile was similar to that observed for IL-6. The IL-17A-treated group was set as the reference at 100% inhibition, whereas the reference drug, secukinumab (400 nM), demonstrated an inhibitory activity of 94.03% (Figure 2B).

### 2.3. Therapeutic Effects of HCB-5300 and HCB-5400 on DSS-Induced Colitis

To investigate the in vivo therapeutic potential of HCB-5300 and HCB-5400, a murine model of acute colitis was established by administering 3% DSS in drinking water for 5 consecutive d (Figure 3A). In addition, colon shortening was attenuated in both the HCB-5300 and HCB-5400 treatment groups. On day 8, the colon length of the DSS-only group (5.0 ± 0.4 cm) was significantly shorter than that of the wild-type group (7.3 ± 0.8 cm, *p* < 0.01). HCB-5300–treated mice showed a slight recovery in colon length compared with the DSS-only group (5.2 ± 0.7 cm), although this difference did not reach statistical significance. In contrast, the HCB-5400 group exhibited a significant increase in colon length to 6.1 ± 0.8 cm (*p* < 0.05; Figure 3C,D). Disease progression was monitored daily by assessing body weight, stool consistency, and rectal bleeding. The DAI scores were markedly lower in both treatment groups than in the DSS-only group (Figure 4).

To place these therapeutic effects in context, we also performed a literature-based benchmark analysis comparing the effects of HCB-5300 and HCB-5400 on body weight, DAI, colon length, and histology with representative data reported for azathioprine, tofacitinib, TAK-875, and the STAT3 inhibitor TTI-101 in DSS-induced colitis models (Appendix A).

### 2.4. HCB Treatment Ameliorates Histological Damage in DSS-Induced Colitis

Histological assessment was performed by scoring mucosal epithelial damage and inflammatory cell infiltration according to the criteria detailed in Table 1. The total histopathological grade was calculated as the sum of the two parameters (range, 0–8). Severe epithelial disruption and massive inflammatory cell infiltration were observed in the DSS-only group, whereas the HCB-5300 and HCB-5400 groups showed relatively preserved colonic architecture with reduced inflammation (Figure 5A).

Notably, the total histopathological grade was significantly reduced in both the HCB-5300 and HCB-5400 treatment groups compared to that in the DSS-only group (Figure 5B). Furthermore, a quantitative analysis of the mucosal damage area, calculated as the percentage of severely damaged epithelium within the total colon area, revealed significant reductions in both treatment groups, with HCB-5400 inducing the most prominent improvement (Figure 5B). Collectively, HCB-5300 and HCB-5400 effectively ameliorated DSS-induced colitis in vivo, with HCB-5400 demonstrating superior therapeutic efficacy.

### 2.5. HCB Treatment Reduces Serum IL-6 Levels in Colitis Mice

The systemic anti-inflammatory effects of HCB compounds were further investigated by measuring the serum levels of the pro-inflammatory cytokine IL-6. Serum IL-6 levels decreased in both treatment groups compared to that of the DSS-only group. HCB-5300 tended to lower IL-6 levels, this change did not reach statistical significance compared with the DSS-only group. Notably, the HCB-5400 group showed a significant reduction in IL-6 levels, underscoring its potent anti-inflammatory activity (Figure 6).

### 2.6. Short-Term HCB Treatment Does Not Alter Bone Microarchitecture

IBD is clinically associated with secondary osteoporosis, as sustained inflammation can impair bone formation and accelerate bone loss. In addition, STAT3 is a key regulator of osteoclast differentiation and has been implicated as a therapeutic target in inflammatory bone loss. Furthermore, our preliminary unpublished data suggested that a lead HCB compound showed a pro-regenerative effect in an ovariectomized (OVX) osteoporosis model. On this basis, we sought to evaluate whether selective STAT3 inhibition by HCB compounds in the IBD model could provide a dual therapeutic benefit by both alleviating intestinal inflammation and improving bone metabolism. The results revealed no significant differences in any of the measured bone parameters between the treatment and control groups (Figure 7). This finding indicates that neither the induction of severe colitis nor the treatments with HCB-5300 and HCB-5400 produced any acute, measurable effects on bone parameters. This is an expected outcome, given that significant changes in bone metabolism typically occur over weeks or months, not within the 8-day duration of this acute model.

## 3. Discussion

Our findings demonstrated that two novel, highly selective STAT3 allosteric inhibitors, HCB-5300 and HCB-5400, show significant anti-inflammatory effects in a DSS-induced murine model of acute colitis, suggesting their potential as a therapeutic strategy for managing acute inflammatory episodes in IBD, particularly UC. Both compounds significantly reduced the hallmark symptoms of colitis, such as body weight loss, DAI, and colon shortening, and showed pronounced reductions in mucosal injury and inflammatory cell infiltration. Notably, HCB-5400 consistently outperformed HCB-5300 in most parameters, indicating promising efficacy.

The central role of STAT3 in IBD pathogenesis is well-established, with its aberrant activation driving the expression of central pro-inflammatory cytokines, such as IL-6 and IL-17 [14,15]. We confirmed in vitro that both HCB-5300 and HCB-5400 potently suppressed IL-6 and IL-17 expression in a dose-dependent manner. Consistent with this finding, our in vivo data showed significant suppression of IL-6, corroborating their effectiveness as STAT3 pathway inhibitors. This targeted action is a key differentiator from that of broader immunosuppressive agents. Considering the persistent need for novel treatments for IBD, due to the limitations in efficacy, safety, and patient response rates of current agents such as corticosteroids, immunomodulators, and biologics [16,17], targeting the STAT3 pathway with high specificity and potency offers a compelling therapeutic strategy [18].

From a clinical translation perspective, these findings are particularly noteworthy. Both HCB-5300 and HCB-5400 exhibited significant anti-inflammatory effects without apparent acute toxicity in vivo. Furthermore, the oral availability and high selectivity of these agents provide practical advantages over currently approved biological therapies, which often require parenteral administration and may be associated with systemic immunosuppression or opportunistic infections [19]. The observed ability of HCB-5400, in particular, to markedly improve the histological and biochemical markers of inflammation indicates its potential as a potent oral treatment for the induction of remission in patients with active IBD. This is a critical therapeutic goal, as IBD often requires long-term treatment where drug safety is paramount [20]. Therefore, the long-term efficacy of these compounds must be validated in chronic and relapsing disease models.

Although this study provides valuable and compelling findings, it nonetheless has several limitations. First, we did not newly assess STAT3 target gene expression by quantitative PCR or p-STAT3 levels by Western blot. Instead, we interpreted the suppression of IL-6 (in both in vivo and in vitro experiments) and IL-17 (in vitro) as an indirect indicator of STAT3 pathway inhibition. In particular, we did not newly assess phospho-STAT3 levels in colonic tissues in vivo and thus relied on cytokine changes and other inflammatory readouts as indirect indicators of STAT3 pathway modulation; future in vivo studies will incorporate Western blot and/or immunohistochemical analyses of phospho-STAT3 and STAT3 target genes in colonic tissue to provide more definitive pharmacodynamic evidence. Therefore, direct evaluation of STAT3 target genes and p-STAT3 under both in vitro and in vivo conditions will be an important goal for future studies. Second, although the ultimate clinical application of these compounds is IBD, our in vitro experiments were conducted primarily in non-intestinal cell lines. Future work will focus on validating the efficacy and mechanism of HCB-5300 and HCB-5400 in human intestinal epithelial cell lines and organoid models, which will better recapitulate the intestinal microenvironment. Third, this study did not include a direct comparison with a standard-of-care positive control group, such as an anti-TNF agent. While our literature-based benchmark analysis provides important context, the absence of an in-study comparator is a limitation that should be considered when evaluating the therapeutic potential of our compounds. Fourth, a comprehensive kinase panel–based off target assessment has not yet been completed at this stage. Nevertheless, several lines of evidence—namely low cytotoxicity, tight concordance between STAT3 inhibition and NF-κB/Th cell modulation, and parallel structure–activity relationships within the HCB series—argue against major off-target–driven effects. A full tyrosine/serine kinase profiling, however, will still be required at the final candidate selection stage.

Moreover, dual inhibition of IL-6 and IL-17 pathways is clinically relevant, as the overproduction of these cytokines correlates with disease severity and IBD progression [21]. Existing treatments targeting these pathways individually have shown mixed success and sometimes unwanted side effects, including the risk of infection and paradoxical exacerbation of the disease, as seen with specific IL-17 inhibitors in CD [22]. Therefore, the ability of HCB-5300 and HCB-5400 to mediate a balanced suppression of both cytokines, as demonstrated in vitro, suggests a potential mechanism that could translate into improved efficacy and tolerability.

Despite these encouraging results, several limitations warrant consideration. This study employed an acute DSS-induced colitis model, which cannot fully recapitulate the chronic, relapsing nature of human IBD [12]. Our investigation into bone metabolism via micro-CT analysis also revealed no significant changes, as confirming bone metabolism effects in such a brief period is rather challenging. However, future analyses in a long-term study model may demonstrate that our compounds also improve bone health, suggesting their potential as dual-effect agents that treat colitis and ameliorate bone loss. Long-term safety, pharmacokinetics, and possible off-target effects, including immunosuppression and organ-specific toxicities, require thorough investigation before clinical application [23]. Additionally, the molecular mechanisms underlying the superior efficacy of HCB-5400 over HCB-5300 need to be elucidated in more detail, along with direct comparisons with existing therapies, such as JAK inhibitors and biologics, in preclinical settings [24].

Future research should focus on evaluating these compounds in chronic and spontaneous IBD models to assess their long-term effects on bone metabolism. Exploring potential combinatorial therapies and expanding toxicological and pharmacodynamic profiles are crucial to support the transition of these compounds to clinical trials [25]. Biomarker studies may further elucidate the subpopulations most likely to benefit from STAT3-targeted approaches. Considering its robust efficacy, oral bioavailability, and mechanistic rationale demonstrated in this study, HCB-5400 emerges as a compelling clinical candidate for IBD treatment, potentially addressing unmet needs in efficacy, route of administration, and safety [26].

In conclusion, our data provide strong evidence for the therapeutic potential of selective STAT3 inhibition in IBD, supporting further preclinical and clinical development of HCB-5300 and, in particular, HCB-5400. If these findings are replicated in more physiologically relevant and chronic models and confirmatory safety data are obtained, these novel compounds may help expand the therapeutic armamentarium for IBD, ultimately improving outcomes for patients with this challenging and burdensome condition.

## 4. Materials and Methods

### 4.1. Animal Model and Ethical Approval

Eight-week-old male C57BL/6J mice (*n* = 20) were purchased from DBL (Eumseong, Republic of Korea) and acclimatized for 1 week before the experiment. All procedures were conducted in accordance with the Animal Protection Act and Act on the Protection and Management of Laboratory Animals, and were approved by the Institutional Animal Care and Use Committee (IACUC) of Woojung Bio Co., Ltd. (Suwon, Gyeonggi-do, Republic of Korea) (Approval No. IACUC2403-034). The mice were housed under specific pathogen-free conditions in Library 3 facility at Woojung Bio Co., Ltd.

### 4.2. Induction of DSS-Induced Colitis and Treatment Administration

Colitis was induced in all groups except the normal control group (DMSO: PEG400:Tween80:0.25% CMC in distilled water; *n* = 5) by replacing drinking water with 3% (*w*/*v*) DSS for 5 consecutive d (days 0–5). On day 5, DSS-containing water was replaced with sterile distilled water and provided ad libitum until the end of the study (day 8).

From the first day of DSS administration, 9-week-old mice in the experimental groups were orally administered either HCB-5300 (25 mg/kg, *n* = 5) or HCB-5400 (12.5 mg/kg, *n* = 5) once daily for 8 d. The negative control group (DMSO: PEG400:Tween80:0.25% CMC in distilled water; *n* = 5) received the same dosing regimen. Disease progression was monitored daily by assessing body weight loss, stool consistency, and rectal bleeding. The DAI score was calculated as the sum of the weight loss, stool consistency, and bleeding scores (0–12 points).

### 4.3. Sample Collection and Processing

At the end of the experiment, the mice were anesthetized using isoflurane (Hana Pharm Co., Ltd., Hwaseong, Gyeonggi-do, Republic of Korea), and whole blood was collected via cardiac puncture. Blood samples were placed into serum separation tubes (BD Biosciences, San Jose, CA, USA) and allowed to clot at room temperature for 30 min, followed by centrifugation at 6797× *g* at 4 °C for 10 min. The resulting serum was aliquoted into 1.5 mL microtubes and stored at −80 °C until further analysis.

Following euthanasia, the entire colon (including the rectum) was excised as a single piece to ensure that no surrounding adipose tissue remained. The colon was carefully straightened and arranged according to the treatment groups for length measurement and photographic documentation. For histopathological analysis, tissues were fixed in 10% neutral-buffered formalin and stored at 4 °C.

### 4.4. Reporter Gene Assays for STAT1 and STAT3

For the STAT1/STAT3 reporter assays, cells were seeded at a density of 2 × 10^4^ cells/well in a 96-well white plate (Greiner, Monroe, NC, USA) and incubated overnight at 37 °C with 5% CO_2_. Then, the cells were treated with either vehicle or the test compounds, and stimulated with 100 ng/mL interferon-gamma (STAT1) or 10 ng/mL IL-6 (STAT3) for 24 h. The cells were lysed, and luciferase activity was measured using the Luciferase Reporter Gene Assay Kit (Promega, Madison, WI, USA) according to the manufacturer’s protocol. The plates were prepared in duplicate: one plate for each luciferase reporter and CellTiter-Glo cell viability assay. Luciferase activity was measured using the luciferase assay reagent (Promega Cat# E1501 for STAT1 and STAT3), and luminescence was detected using a microplate luminometer (BioTek, Winooski, VT, USA). For data analysis, the average background luminescence was subtracted from all well readings. Next, the fold induction of STAT luciferase reporter expression was calculated by dividing the background-subtracted luminescence of stimulated wells by the average background-subtracted luminescence of unstimulated control wells.

### 4.5. Histological Analysis and Immunohistochemical Staining

Paraffin blocks were prepared from large intestinal segment tissues, sectioned, and stained with hematoxylin and eosin. The severity of colitis-induced mucosal epithelial damage and inflammatory cell infiltration was evaluated using a scoring system, and the sum of these scores determined the final histopathological grade (0–8). Additionally, the percentage of severely damaged epithelial area within the total colon area was quantified.

### 4.6. Enzyme-Linked Immunosorbent Assay

Serum samples preserved post-sacrifice were analyzed for IL-6 levels using an enzyme-linked immunosorbent assay (ELISA), performed according to the manufacturer’s instructions (Mouse IL-6 ELISA Kit, Invitrogen, Carlsbad, CA, USA, Cat# KMC0061).

### 4.7. Evaluation of Anti-Inflammatory Activity (IL-6 and IL-17 Inhibitory Assays)

RAW264.7 and HaCaT cells were seeded in 96-well plates (2 × 10^4^ and 1 × 10^4^ cells/well, respectively) and incubated for 24 h. Cells were pre-treated with various concentrations of HCB compounds or reference drugs (dexamethasone for IL-6, secukinumab for IL-17) for 1 h, followed by stimulation with LPS (100 ng/mL, RAW264.7) or IL-17A (100 ng/mL, HaCaT) for 48 h. Supernatants were collected, and IL-6 or IL-17 levels were measured using ELISA (R&D Systems, Minneapolis, MN, USA) according to the manufacturer’s protocol. Absorbance was determined at 450 nm using an EnVision microplate reader (PerkinElmer, Waltham, MA, USA).

### 4.8. Statistical Analysis

All data are presented as mean ± standard deviation. The statistical significance of in vivo data between groups was assessed using the Mann–Whitney U test for non-parametric data. Statistical analyses were performed using IBM SPSS Statistics software (version 25.0; IBM, Armonk, NY, USA) on a Windows operating system. A *p* value < 0.05 was considered statistically significant.

## Figures and Tables

**Figure 1 ijms-26-11981-f001:**
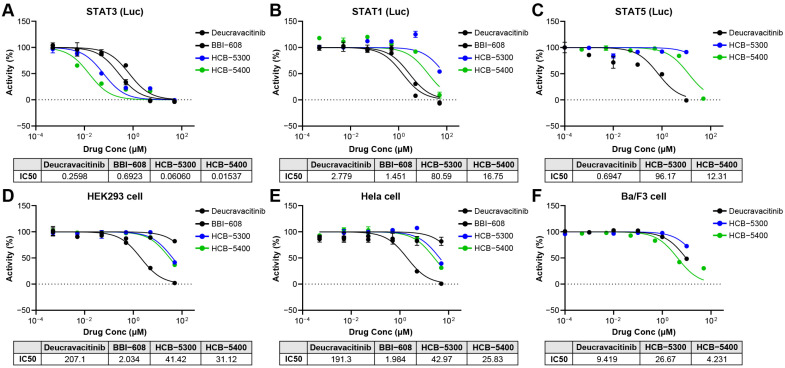
HCB compounds selectively inhibit STAT3 transcriptional activity without affecting cell viability. (**A**–**C**) Dose-dependent inhibition of STAT1 (**A**), STAT3 (**B**) and STAT5 (**C**) transcriptional activity by HCB-5300, HCB-5400, and deucravacitinib. The activity of each STAT isoform was measured using a luciferase reporter assay, and IC50 values were calculated. (**D**–**F**) Viability of HeLa (**D**), HEK293 (**E**) and Ba/F3 (**F**) cells after treatment with various concentrations of HCB compounds. Cell viability was measured in parallel with the assessment of STAT3 activity inhibition to confirm the absence of cytotoxicity from the compounds. Data are presented as median (interquartile range [IQR]). Statistical significance was assessed on day 7 data using two-tailed Mann–Whitney U tests for pairwise comparisons versus the 3% DSS group. Groups: wild type (WT), 3% DSS, 3% DSS + HCB-5300 (25 mg/kg), 3% DSS + HCB-5400 (12.5 mg/kg); *n* = 5 per group.

**Figure 2 ijms-26-11981-f002:**
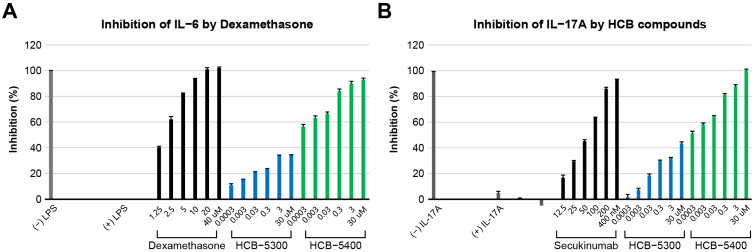
Evaluation of the anti-inflammatory activity of HCB compounds. (**A**) Inhibitory effect on interleukin (IL)-6 production. RAW264.7 cells were treated with HCB compounds at six different concentrations (0.0003–30 μM), and inhibitory activity on lipopolysaccharide (LPS)-induced IL-6 expression was measured. The results are presented as the percentage of inhibition relative to the LPS-treated control group. (**B**) Inhibitory effect on IL-17 production. HaCaT cells were treated with HCB compounds at six different concentrations (0.0003–30 μM), and inhibitory activity on IL-17A-induced expression was measured. The results are presented as the percentage of inhibition relative to the IL-17A-treated control group.

**Figure 3 ijms-26-11981-f003:**
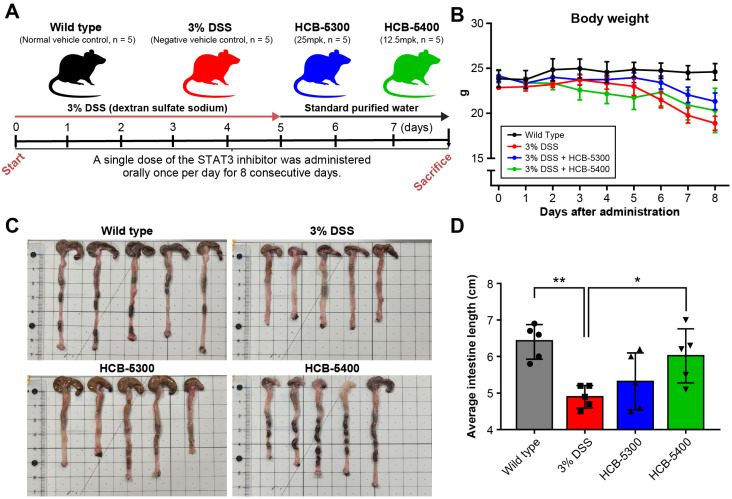
Study design and the changes in body weight and whole large intestine lengths. (**A**) Schematic of the dextran sulfate sodium (DSS)-induced colitis model and dosing regimen: C57BL/6 mice received 3% DSS; HCB-5300 (25 mpk) and HCB-5400 (12.5 mpk) were administered orally for 7 days. (**B**) Absolute change in body weight, recorded daily from days 0 to 8. (**C**) Macroscopic appearance of the whole intestine from mice sacrificed on day 7. (**D**) The graph illustrates the total lengths of the large intestines under specific experimental conditions. Data are presented as median (interquartile range [IQR]). Statistical significance was assessed on day 7 data using two-tailed Mann–Whitney U tests for pairwise comparisons versus the 3% DSS group; * *p* < 0.05, ** *p* < 0.01. The groups are represented by symbols: (●) Wild type, (■) 3% DSS, (▲) 3% DSS + HCB-5300 (25 mg/kg), and (▼) 3% DSS + HCB-5400 (12.5 mg/kg); *n* = 5 per group.

**Figure 4 ijms-26-11981-f004:**
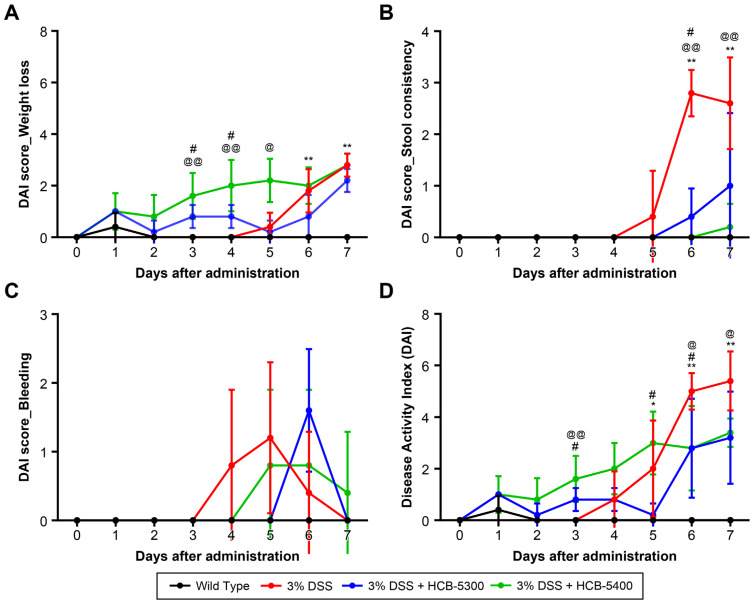
Disease Activity Index (DAI) and its individual components during the experimental period. (A) Weight loss score; (B) Stool consistency score; (C) Bleeding score; and (D) the overall DAI score. Data are presented as median (interquartile range [IQR]). Statistical significance was assessed on the indicated days’ data using two-tailed Mann–Whitney U tests for pairwise comparisons. Significance is denoted as follows: * *p* < 0.05, ** *p* < 0.01 (vs. Wild type); # *p* < 0.05 (HCB-5300 vs. 3% DSS); @ *p* < 0.05, @@ *p* < 0.01 (HCB-5400 vs. 3% DSS).

**Figure 5 ijms-26-11981-f005:**
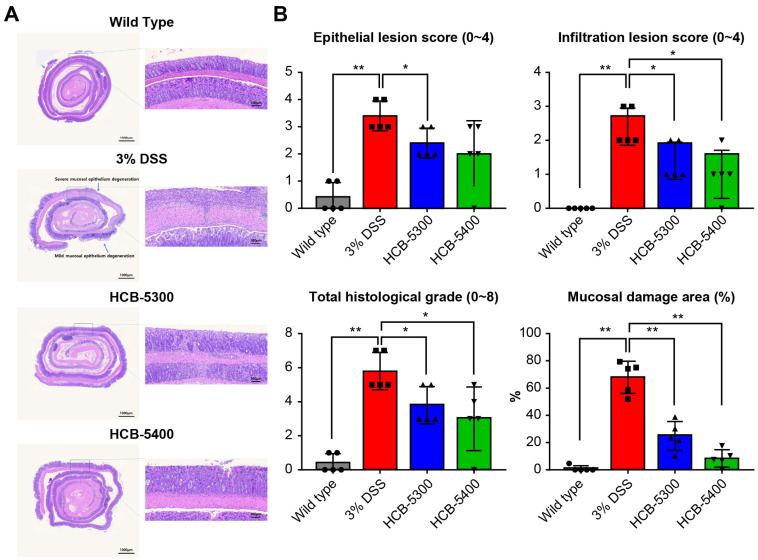
Histopathological changes in the colon. (**A**) Representative H&E-stained Swiss-rolled colon sections (scale bar = 100 μm). (**B**) Quantification of epithelial damage, inflammatory cell infiltration, damaged mucosal epithelial area (%), and total histological grade on day 7. Data are presented as median (interquartile range [IQR]). Statistical significance was assessed on day 7 data using two-tailed Mann–Whitney U tests for pairwise comparisons versus the 3% DSS group; asterisks denote significance: * *p* < 0.05, ** *p* < 0.01. The groups are represented by symbols: (●) Wild type, (■) 3% DSS, (▲) 3% DSS + HCB-5300 (25 mg/kg), and (▼) 3% DSS + HCB-5400 (12.5 mg/kg); *n* = 5 per group.

**Figure 6 ijms-26-11981-f006:**
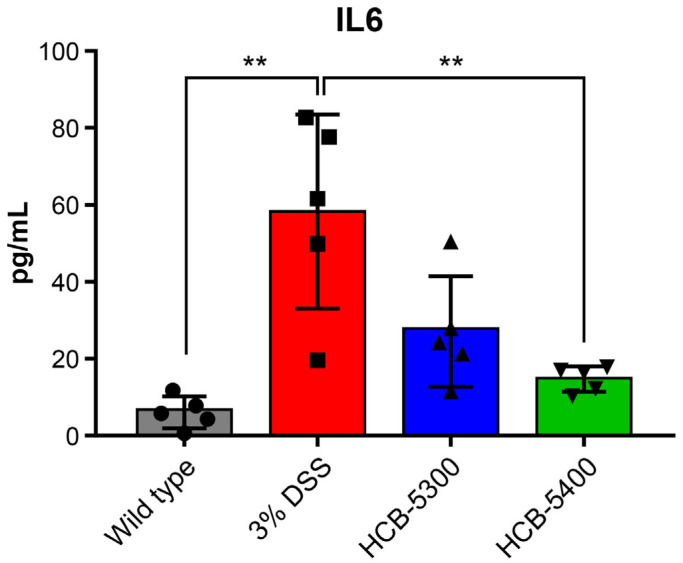
Serum levels of IL-6 on day 7. Serum IL-6 levels were measured on day 7 using enzyme-linked immunosorbent assay. Data are presented as median (interquartile range [IQR]). Statistical significance was assessed on day 7 data using two-tailed Mann–Whitney U tests for pairwise comparisons versus the 3% DSS group; asterisks denote significance: ** *p* < 0.01. The groups are represented by symbols: (●) Wild type, (■) 3% DSS, (▲) 3% DSS + HCB-5300 (25 mg/kg), and (▼) 3% DSS + HCB-5400 (12.5 mg/kg); *n* = 5 per group.

**Figure 7 ijms-26-11981-f007:**
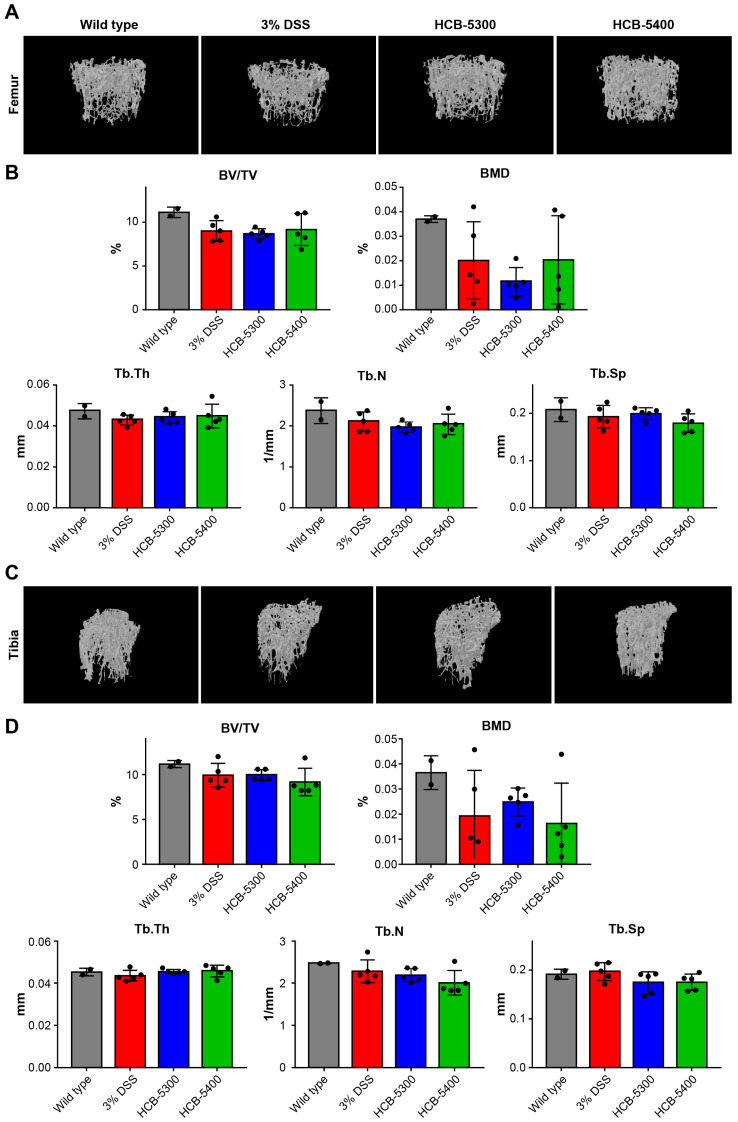
Micro-computed tomography analysis of the femur and tibia. (**A**) Three-dimensional (3D) image of the femur. (**B**) Analysis of BV/TV, BMD, and trabecular structure of the femur. (**C**) 3D image of the tibia. (**D**) Analysis of BV/TV, BMD, and trabecular structure of the tibia. Data are presented as median (interquartile range [IQR]). Statistical significance was assessed on day 7 data using two-tailed Mann–Whitney U tests for pairwise comparisons versus the 3% DSS group. The groups are represented by symbols: (●) Wild type, (■) 3% DSS, (▲) 3% DSS + HCB-5300 (25 mg/kg), and (▼) 3% DSS + HCB-5400 (12.5 mg/kg); *n* = 5 per group.

**Table 1 ijms-26-11981-t001:** Histologic scoring criteria for colonic injury. The total histologic score (range: 0–8) is the sum of the epithelial injury and inflammatory cell infiltration scores. All tissues were stained with H&E, and scoring was performed on randomly selected fields by observers blinded to the experimental groups.

	Epithelial Lesion Score	Infiltration Lesion Score
score	Degree of Epithelial damage	Degree of Infiltration
0	normal	none
1	loss of goblet cells	infiltration surrounding crypt basis
2	loss of goblet cells in large areas	infiltration into the mucosa
3	loss of crypts	widespread infiltration into the mucosa
4	loss of crypts in large areas	infiltration into the submucosa

## Data Availability

The original contributions presented in this study are included in the article/Appendix A. Further inquiries can be directed to the corresponding author.

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
