# Peer review of "Selective STAT3 Allosteric Inhibitors HCB-5300 and HCB-5400 Alleviate Dextran Sulfate Sodium-Induced Ulcerative Colitis in Mice"

_ijms, 2025, doi:10.3390/ijms262411981_

Round 1

Reviewer 1 Report

Comments and Suggestions for Authors

The study explores the therapeutic potential of two novel selective STAT3 allosteric inhibitors, HCB-5300 and HCB-5400, in a DSS-induced mouse model of ulcerative colitis. The abstract and introduction are generally clear and well-structured, effectively highlighting the key findings and the relevance to IBD therapy. The use of multiple endpoints—including disease activity index, colon length, histopathology, and IL-6 levels—strengthens the evidence for STAT3-mediated anti-inflammatory effects. However, several aspects of the results require clarification or reorganization. Therefore, I recommend the manuscript for publication after the authors have addressed the following points. My detailed comments are as follows:

  1. The authors should provide more background on the HCB molecules, including their origin, chemical structure, and any relevant previous studies.
  2. While IL-6 reduction is presented as a marker of STAT3 inhibition, it would be helpful to clarify whether downstream STAT3 target genes were assessed directly to confirm pathway inhibition, for example, via qPCR or Western blot analyses.
  3. Although the binding affinity and anti-inflammatory effects of the HCB molecules have been demonstrated in multiple cell lines, since the application is intended for IBD, their effects on intestinal-derived cell lines should also be investigated.
  4. The statistical significance between HCB-5300 and the IBD model is missing in Figures 3 and 6; the authors should revise the figures to include this information.
  5. The rationale for investigating bone metabolism is unclear, and the presentation of these results is confusing. The authors should clarify the aim and logical connection to the main study.

Reviewer 2 Report

Comments and Suggestions for Authors

The manuscript “Selective STAT3 Allosteric Inhibitors HCB-5300 and HCB-5400 Alleviate Dextran Sulfate Sodium-Induced Ulcerative Colitis in Mice” investigates two novel selective allosteric STAT3 inhibitors, evaluating their in vitro activity and therapeutic efficacy in a murine model of DSS-induced colitis. Based on ICâ‚…â‚€ data, cytokine inhibition assays, and histopathological analyses, the authors show that both compounds reduce inflammation, preserve colon integrity, and decrease systemic IL-6 production, with HCB-5400 being consistently more effective. From the perspective of enzyme modeling and pharmacodynamics, the results suggest that allosteric selectivity for STAT3 is functionally relevant and translates into a robust anti-inflammatory response, supporting the potential of the compounds as therapeutic candidates for IBD.

The Introduction adequately presents the clinical relevance (lines 38–87), justifying the targeting of STAT3 and the need for selective inhibitors. The references are up-to-date and adequate (2021–2024), covering essential literature on IBD, DSS, and STAT3.

The Abstract does not mention study limitations—slight underexposure (lines 16–33)—and the study objective could be more explicit—the hypothesis is implicit but not clearly stated (lines 75–86).

Regarding the research question and relevance, it is valid and well-defined, even if the mechanistic rationale for the differences between HCB-5300 and HCB-5400 is not fully developed (lines 274–276).

In methodological terms, at a general level, the article is well described: i) DSS model adequately described and with appropriate citations (lines 68–74; 302–306); ii) Dosages and routes of administration are described (lines 307–310). iii) Animal ethics complied with and approved (lines 294–301); iv) Sufficient detail for reproducibility of ELISA, luciferase, and histology assays (lines 326–359).

However, some points for improvement can be identified and are itemized below:

#1 Absence of a positive in vivo control group (e.g., dexamethasone, anti-TNF): There is no direct comparison with standard IBD therapies (lines 155–175; 219–257).

#2 Justification for doses based on ICâ‚…â‚€ and PK mentioned, but not described (lines 108–111). Lack of description of bioavailability, half-life, or numerical rationale for dose.

#3 Lack of assessment of relevant off-targets via kinetic assays or tyrosine/serine kinase panels.

#4 Temporal heterogeneity: DSS for 5 days, but treatment for 8 is not fully explained (lines 307–313).

#5 Lack of direct pharmacodynamic assay to demonstrate STAT3 inhibition in vivo (phospho-STAT3 in tissues, Western blot or IHC). This is a relevant gap.

In the statistical evaluation, some inconsistencies can also be identified:

#6 No correction test for multiple comparisons.

#7 ANOVA + Mann–Whitney mixture is methodologically inconsistent for small samples (lines 361–367).

#8 Information on normality or variance is lacking, even to justify ANOVA.

Regarding the RESULTS and respective CONCLUSIONS, it is suggested that the authors rectify the writing in some aspects, which are itemized below:

RC#1 General conclusions too strong for the acute DSS model. The authors repeatedly state that the compound has clinical potential (lines 247–285), but the model does not recapitulate chronicity, and it has not been tested in TNBS, IL-10-KO, or chronic models. They should qualify the translational validity.

RC#2 Absence of pharmacokinetic data despite being reported as a basis for dose (lines 108–111). Logical flaw: conclusions about the “optimal” dose cannot be supported.

RC#3 Lack of demonstration of STAT3 inhibition in vivo. No measurement of phosphorylated STAT3 in colonic tissue (Western/IHC). No direct evidence that the anti-inflammatory effect is due to the STAT3 pathway and not to off-target pathways. This is the biggest scientific gap.

RC#4 L-17 in vivo is not measured (lines 149–152), but is referred to in the discussion as if it were confirmed. Extrapolated deduction.

RC#5 Differences between HCB-5300 and HCB-5400 are not explained mechanistically. No structural, kinetic, binding profile or affinity data.

RC#6 Bone metabolism (lines 211–218). Conclusion is technically correct (“no changes”), but the rationale for including this data is not clearly justified. Too short a time for meaningful conclusions.

Round 2

Reviewer 1 Report

Comments and Suggestions for Authors

I agree to publish this work in its present form. 

Reviewer 2 Report

Comments and Suggestions for Authors

The effort by the authors to address the comments of The Peer-Review Report is recognized and congratulated! Thank you very much!